# Estimating the Prevalence of De Novo Monogenic Neurodevelopmental Disorders from Large Cohort Studies

**DOI:** 10.3390/biomedicines10112865

**Published:** 2022-11-09

**Authors:** Madelyn A. Gillentine, Tianyun Wang, Evan E. Eichler

**Affiliations:** 1Department of Laboratories, Seattle Children’s Hospital, Seattle, WA 98105, USA; 2Department of Medical Genetics, Center for Medical Genetics, School of Basic Medical Sciences, Peking University Health Science Center, Beijing 100191, China; 3Key Laboratory for Neuroscience, Neuroscience Research Institute, Peking University, Ministry of Education of China & National Health Commission of China, Beijing 100191, China; 4Department of Genome Sciences, University of Washington School of Medicine, Seattle, WA 98105, USA; 5Howard Hughes Medical Institute, University of Washington, Seattle, WA 98195, USA

**Keywords:** neurodevelopmental disorders, rare disease, de novo, monogenic, prevalence

## Abstract

Rare diseases impact up to 400 million individuals globally. Of the thousands of known rare diseases, many are rare neurodevelopmental disorders (RNDDs) impacting children. RNDDs have proven to be difficult to assess epidemiologically for several reasons. The rarity of them makes it difficult to observe them in the population, there is clinical overlap among many disorders, making it difficult to assess the prevalence without genetic testing, and data have yet to be available to have accurate counts of cases. Here, we utilized large sequencing cohorts of individuals with rare, de novo monogenic disorders to estimate the prevalence of variation in over 11,000 genes among cohorts with developmental delay, autism spectrum disorder, and/or epilepsy. We found that the prevalence of many RNDDs is positively correlated to the previously estimated incidence. We identified the most often mutated genes among neurodevelopmental disorders broadly, as well as developmental delay and autism spectrum disorder independently. Finally, we assessed if social media group member numbers may be a valuable way to estimate prevalence. These data are critical for individuals and families impacted by these RNDDs, clinicians and geneticists in their understanding of how common diseases are, and for researchers to potentially prioritize research into particular genes or gene sets.

## 1. Introduction

Rare diseases, in particular rare neurodevelopmental disorders (RNDDs), have proven to be challenging to understand epidemiologically. There are several definitions of “rare disease” that vary globally [1,2]. The current definition of a rare disease in the United States is a disease that impacts fewer than 200,000 individuals, or approximately 86 per 100,000 individuals at the time the American Orphan Drug Act was passed in 1983. Other global definitions range from 5 to 76 per 100,000 individuals. Overall, an estimated 3.5–5.9% of the global population has a rare disease, many of which are RNDDs mostly diagnosed in early childhood [3].

Neurodevelopmental disorders (NDDs), impacting up to 17% of the population, are a clinically and genetically heterogenous group of diagnoses [4]. NDDs as a whole are not rare; but each individual RNDD with known genetic cause only accounts for 1% or less of NDD cases. Many studies have shown that de novo variants (DNVs) are key contributors to such disorders [5,6,7,8,9]. The prevalence of these disorders is key for families looking for community, researchers, clinicians, and in pharmaceutical development [10].

Due to the scarcity of these RNDDs, with variable expressivity and incomplete penetrance, traditional epidemiological methods are challenging to assess. Additionally, many monogenic RNDDs share clinical features or lack pathognomonic features, making it difficult to identify them without genetic testing. Another challenge is the barriers to genetic testing resulting in underdiagnoses of many RNDDs, which leave patients uncounted.

Multiple approaches have been taken to understand the prevalence and/or incidence of RNDDs. Clinical data have been utilized for deletion/duplication syndromes mediated by nonallelic homologous recombination [11]. The number of published articles has also been used as a potential metric for prevalence [12]. For monogenic disorders, Nguengang Wakap et al. (2020) utilized point prevalence (number of cases in the population at one time/total population at the same time point), although not by gene but by inheritance pattern [3]. The incidence of de novo monogenic RNDDs has been elegantly estimated by López-Rivera et al. (2020), utilizing mutational constraint and probability of mutation to estimate based on mutation rate of individual genes [13,14]. Additionally, several resources report estimated prevalence, such as Orphanet, the National Organization for Rare Disorders (NORD), and others, although it is not always clear how these numbers are determined.

In order to assess the prevalence of autosomal dominant de novo monogenic RNDDs, we utilized the DNV data from multiple large cohorts of individuals with NDDs, specifically developmental delay/intellectual disability (DD/ID), autism spectrum disorder (ASD), and epilepsy. Cohorts include the Deciphering Developmental Disorders studies, Autism Sequencing Consortium, Simons Simplex Collection, SPARK, and MSSNG [5,6,7,15,16,17,18,19,20,21,22,23,24,25,26,27,28]. It is likely that these large studies of DNV provide the most comprehensive counts available of individuals with specific neurodevelopmental-related genetic alterations. From these cohorts (n = 50,377), we estimated the prevalence of variation of over 11,000 genes with reported variation in NDDs among the general population, which is positively correlated to the previously estimated incidence. We also identified the most often mutated genes among NDDs broadly, DD/ID and ASD. Finally, we determined that social media group member numbers may be a valuable way to estimate prevalence. These data are critical for individuals and families impacted by these rare disorders, clinicians and geneticists in their understanding of how common diseases are, and researchers to potentially prioritize research into particular genes or gene sets.

## 2. Methods

### 2.1. Cohorts and Samples

Published data were utilized from genome and exome studies (Appendix A). Cohorts included studies focusing on NDDs (n = 50,377), ASD (n = 16,125), DD/ID (n = 31,191), and epilepsy (n = 1389). Utilizing published data, we avoided double counting probands that were in multiple studies to the best of our ability [26]. A subset of variants was Sanger validated in their original studies with greater than 90% of variants being confirmed, suggesting that any false positives on prevalence estimates would be negligible. Phenotypic and diagnostic information varies by cohort but typically included ASD diagnoses by both the ADOS and ADI-R, cognitive testing, Diagnostic and Statistical Manual of Mental Disorders (DSM) diagnoses (mostly DSM-V, although some studies were performed before its release in 2013), and basic medical screening (Appendix A) [29,30,31,32,33,34,35,36,37,38,39,40,41,42,43,44,45].

### 2.2. Prevalence Estimation

Prevalence information for ASD, DD, and epilepsy was used from Zablotsky and Black (2020) to comprise our NDD prevalence (Table 1) [4]. While NDDs as a whole affect 17% of 3- to 17-year-old children in the US, we focused on those that were well represented in our de novo NDD cohort. Coding, nonsynonymous variant counts were computed from each study (Appendix A). The number of variants in each gene was normalized by the observed/expected values for each type of variant obtained from gnomAD v2.1.1 (Appendix A). Genes with negative values resulting from normalization or no constraint metrics available were excluded. The proportion of cases in our combined cohort was multiplied by the estimated prevalence of RNDDs and extrapolated to the prevalence in 100,000 individuals.

Estimates were performed for NDDs, DD/ID, and ASD independently. The number of probands for epilepsy was dramatically lower than DD/ID and ASD; thus, this was not calculated separately due to an inaccurate representation of cases of epilepsy. Variants were also separated by variant type: all DNVs, de novo likely gene disrupting (dnLGD) variants, de novo missense (dnMIS) variants, and de novo severe missense variants with a CADD score greater than or equal to 30 (dnMIS30). Candidate NDD genes were assessed separately and determined by combining statistically significant genes from multiple large cohort studies (n = 468) [7,8,9,26] (Appendix A).

### 2.3. Comparison to Previous Incidence Estimates

Our estimates were compared to birth incidence rates from López-Rivera et al. (2020) using Pearson’s correlations in R Studio (2022.02.2-485, R version 4.2.0). Correlation analyses were performed in R for the gene level and cohort level. For the gene level, the number of DNVs was rounded to the nearest integer. Then, Fisher exact tests between genes that were reported in both our cohort and in that of López-Rivera et al. (2020) were performed in R with Bonferroni correction accounting for all genes (n = 20,000) and number of probands tested (n = 50,377). The 11,461 genes analyzed all had DNVs in our cohort, while the remaining genes in the genome did not in the data used. As previous estimates were not calculated by phenotype, our analysis was only performed for the total NDD cohort.

### 2.4. Comparison to Social Media Group Numbers for Top NDD Genes

We searched Facebook for each gene name and/or known disorder for the top 500 genes as well as any gene that had an OMIM disease entry (n = 294 genes with Facebook groups, Appendix A). The number of members in each group was compared to the estimated prevalence using Pearson’s correlations.

## 3. Results

### 3.1. Prevalence Estimates for All NDDs

We assessed the number of cases in our total NDD cohort for each gene with at least one variant. The number of variants was normalized by observed/expected counts obtained from gnomAD v2.1.1 for dnLGD and dnMIS variants. The dnLGD and dnMIS variants were summed to estimate all DNV prevalence. Utilizing the prevalence estimates from Zablotsky and Black (2020), we calculated prevalence among individuals with NDDs and the prevalence in the general population.

All genes examined met the criteria for rare disease. The most often mutated gene in our NDD cohort was *ARID1B*, accounting for 0.3% of all DNVs as well as the highest proportion of dnLGD variants (dnLGD = 0.25%, dnMIS = 0.05%, dnMIS30 = 0.02%) (Figure 1A,B, Table 2 and Appendix A). This resulted in a prevalence of *ARID1B* variants of 11.1/100,000 individuals (95% CI: 9.9–12.3/100,000 individuals; 1/9009 individuals (95% CI: 1/10,136–8109 individuals)). An *ARID1B*-related disorder is typically due to loss-of-function variants; so, the dnLGD prevalence may be more accurate (10.7/100,000 individuals; 95% CI: 9.5–11.9/100,000 individuals (1/9372 individuals; 95% CI: 1/10,543–8435 individuals)), although missense variants have been reported [46]. This estimate is similar to previous estimates (Table 3) [47].

The gene with the highest proportion of missense variants identified in our NDD cohort was *DDX3X*, accounting for 0.14% of all dnMIS variants (DNV = 0.3% dnLGD = 0.11%, dnMIS30 = 0.06%) (Figure 1C, Table 2 and Appendix A). This resulted in a prevalence of *DDX3X*-related NDD of 9.3/100,000 individuals (95% CI: 8.2–9.2/100,000 individuals (1/10,798 individuals; 95% CI: 1/12,147–10,824)). Previous estimates of *DDX3X*-related NDD were 1–3% of DD/ID in females, suggesting this may be an under-ascertained group in our cohort, although our cohort also had individuals without DD/ID diagnoses [62]. The *DYNC1H1* gene had the most severe missense variants (MIS30) in our NDD cohort, accounting for 0.1% of all variants (DNV = 0.16%, dnLGD = 0.1%, dnMIS = 0.16%) (Figure 1D, Appendix A). While still rare, this suggests that *DYNC1H1* variants may be under-recognized in NDD cohorts [63].

### 3.2. Prevalence Estimates for DD/ID

Cohorts in which the primary diagnosis was DD or ID were analyzed separately. In general, the pattern was similar to the entire NDD cohort, likely due to the larger DD sample size. The gene most often mutated in DD was *ARID1B*, accounting for 0.38% (dnLGD = 0.3%, dnMIS = 0.04%, dnMIS30 = 0.02%) of all DNVs (Figure 2A, Table 4 and Appendix A). *ARID1B* was also the most frequently mutated in dnLGD variants (Figure 2B). This resulted in the frequency of an *ARID1B*-related disease with DD of 4/100,000 individuals (95% CI: 3.7–4.7/100,000 individuals (1/24,816 individuals; 95% CI: 1/27,013–21,225)).

The gene with the highest proportion of missense variants identified was *DDX3X*, accounting for 0.2% of all dnMIS variants (DNVs: 0.37%, dnLGD = 0.13 %, dnMIS = 0.2%, dnMIS30 = 0.09%) (Figure 2C, Table 4 and Appendix A). *DYNC1H1* had the highest percentage of severe missense (dnMIS30) variants (DNV: 0.2%, dnLGD: 0.003%, dnMIS: 0.2%, dnMIS30: 0.13%). This resulted in a prevalence of *DDX3X*-related NDD with DD of 3.6/100,000 individuals (95% CI: 3.5–4.4/100,000 individuals (1/27,610 individuals; 95% CI: 1/28,916–22,720). *DYNC1H1* had the most severe missense variants in our DD cohort, accounting for 0.13% of all variants (DNV = 0.2%, dnLGD = 0.003%, dnMIS = 0.05%) (Figure 2D).

### 3.3. Prevalence Estimates for ASD

Cohorts in which the primary diagnosis was ASD were analyzed separately. Notably, most genes had similar variant numbers between the DD and ASD cohorts, but not necessarily the same ranking. The gene most often mutated in ASD was *SCN2A*, accounting for 0.22% of all DNVs (dnLGD = 0.01%, dnMIS = 0.12%, dnMIS30 = 0.5%) (Figure 3A, Table 5 and Appendix A). *SCN2A* also accounted for the highest prevalence of dnMIS and dnMIS30 variants (Figure 3C,D). This is consistent with previous ASD meta-analyses [26]. This resulted in a prevalence of an *SCN2A*-related disorder with ASD of 4/100,000 individuals (95% CI: 3.6–4.4/100,000 individuals (1/24,626 individuals; 95% CI: 1/27,984–22,801)).

For dnLGD variants, the most often mutated gene in ASD was *ADNP*, accounting for 0.12% of all dnLGD variants (DNVs: 0.16%, dnMIS: 0.03%, dnMIS30: 0%) (Figure 3B). This resulted in a prevalence of an *ADNP*-related disorder with ASD of 3/100,000 individuals (95% CI: 2.7–3.3/100,000 individuals (1/33,107; 95% CI: 1/37,622–30,655)).

### 3.4. Comparison to Previous Estimates

López-Rivera et al. (2020) estimated the incidence for 100 known monogenic disorders as well as the mutation incidence of over 1000 variation intolerant genes. We compared our estimates to theirs using correlation analysis. All variant categories’ prevalence was significantly positively correlated with previous incidence estimates (Figure 4, Appendix A).

For all DNVs in NDDs, there was a significant positive pairwise correlation between the incidence from López-Rivera et al. (Pearson’s correlation coefficient (PCC) = 0.51 (95% CI: 0.5–0.53, *p* < 0.0001)) (Figure 4A). The dnLGD and dnMIS variants for all NDDs were also significantly positively correlated to the López-Rivera et al. estimates (PCC = 0.3: *p* > 0.0001 (95% CI: 0.26–0.3) and PCC = 0.6, *p* < 0.0001 (95% CI: 0.6–0.63)), respectively (Figure 4B,C). For all DNVs and dnMIS variants, NDD candidate genes’ prevalence was significantly correlated with previous incidence estimates (Appendix A). No genes had a significantly different prevalence of mutation when using Bonferroni or FDR correction.

Most genes (n = 6681) had a higher prevalence than previous incidence estimates, as expected since prevalence accounts for all cases and incidence is cases in a year. However, some of these genes may also have been over-ascertained in our cohort (n = 468 NDD candidate genes). Genes with a lower prevalence than incidence (n = 1056; 249 NDD candidate genes) may have had lethal phenotypes or have been under-ascertained in our cohort. The proportion of NDD candidate genes among genes with lower prevalence than incidence (19%) was significantly higher than genes with higher prevalence than incidence (1.5%, Chi squared test, *p* = 0.0005, Appendix A). No genes showed significantly different mutation prevalence after Bonferroni correction.

### 3.5. Comparison to Social Media Groups

One potential estimate of how many individuals and families may be affected by these monogenic disorders is through their social media groups, i.e., how many members does a group have. This likely represents parents of children with rare disorders, and mostly mothers [10]. Over 4000 pediatric rare diseases have Facebook support groups. While membership is limited by computer and internet access, as well as interest in connecting with other families, this may be a reasonable metric for prevalence of these disorders.

To assess this, we found Facebook groups for the top 500 genes and any genes that had a named disorder (n = 294 genes with Facebook groups) (Appendix A). Foundation pages were not included, and the group with the highest number of members was used. Gene and syndrome names were used to identify Facebook groups.

The number of Facebook group members was positively correlated with prevalence (PCC = 0.31) (Appendix A). This moderate correlation suggests that there is an underdiagnosis for many of these monogenic de novo disorders. Interestingly, 66 of the 293 genes analyzed were not significantly enriched among NDD meta-analyses.

## 4. Discussion

The prevalence of most monogenic RNDDs has yet to be determined, and those with estimates are often anecdotal. An accurate estimate of the prevalence is important in understanding each disorder, which also has an impact on research funding and focus. Additionally, there is value in individuals being counted in rare disease [64]. Recently, it has been suggested that there are over 11,000 individual rare diseases, a number that is likely to increase. By identifying individuals with each disorder and determining their prevalence, we can better contribute to our knowledge of rare disease. In combination with cohort-based estimates, incidence estimates from mutation rates, and social media analysis, we hope to have a more comprehensive understanding of the prevalence of these rare disorders.

Utilizing the collection of probands from large sequencing studies that best represent multiple NDD-affected populations to date, we showed the prevalence of de novo variation among NDDs broadly, which, in our cohort, included DD/ID, ASD, epilepsy, and other diagnoses (Figure 1). Our results showed that while most monogenic RNDDs are likely underdiagnosed based on prevalence estimates, they also each account for fewer individuals with NDDs than previously thought. Often, it is reported that each NDD candidate gene accounts for less than 1% of the individuals diagnosed. Here, we showed that each gene accounts for even fewer individuals, with the highest percentage being 0.3% of individuals with NDDs for *ARID1B* (Figure 1A, Table 2 and Appendix A). The GeneReviews for Coffin-Siris syndrome (CSS), of which ~37% of cases are due to *ARID1B* variants, reports that fewer than 200 individuals with CSS have been identified, although a literature and social media review suggests that this number is higher [65,66,67]. Our results suggest there is a considerable underdiagnosis of this syndrome, and this pattern is likely the same for other genetic RNDDs.

Previous studies have tried to use novel methods to determine the prevalence, including using mutation rates and number of papers published [12]. In a similar vein, we compared number of members in social media groups with prevalence estimates (Appendix A). While not significant, there is a positive correlation between number of Facebook group members of a rare disease group and the prevalence of that rare disease. Those with higher estimated prevalence but lower numbers of Facebook group members may represent underdiagnosed or misdiagnosed disorders.

While positively correlated, there are notable differences between our prevalence estimates and previous prevalence or incidence estimates. To an extent, we expect prevalence to be higher than incidence, as incidence is the number of new cases per year, and this is the case for many genes. Several genes are overrepresented compared to their estimated incidence, suggesting possible ascertainment bias. In contrast, many genes have markedly decreased prevalence compared to the estimated incidence, which could be due to a range of factors. We only focused on DNVs, and some of these monogenic disorders have carrier parents, affected or unaffected. Given our DNV-only focus, our cohort likely will have higher accuracy for more severe conditions. We also assumed 100% penetrance for our calculations. It is likely that there are variants that are not fully penetrant or result in subclinical features; thus, those probands may not have been included in our cohort. We also did not consider mortality, which may decrease the prevalence, although most of these disorders are not perinatal lethal. However, the few disorders that are perinatal lethal, such as *MECP2* variants in males, combined with decreased lifespan of individuals with NDDs (average age ~60 years of age) may contribute to the prevalence and be absent from our calculations [68]. Additionally, we only discerned dnLGD and dnMIS or dnMIS30 variants. This leads to some inaccuracy, as there are syndromes that are caused by neither of these variant types but were analyzed in our cohort. Some genes appear to have had a much higher prevalence in our cohort versus the incidence in López-Rivera et al.’s analysis but are skewed due to mutation mechanisms, such as *PPM1D* or *ADNP*, both of which are causative for disease by nonsense and frameshift variants in the penultimate exon that result in truncated proteins escaping nonsense mediated decay. Additionally, there are genes in both the López-Rivera et al., 2020, estimates and ours that are not pathogenic, such as *TTN,* that may skew our correlations, although our normalization with constraint measures aimed to avoid such issues. Furthermore, there are genes that we know to be pathogenic that may have better estimates based on mutation rate than our cohort due to the rarity of these syndromes. Such genes highlight our ascertainment bias, with disorders that have a higher frequency of ASD and/or DD/ID having better estimates. These include disorders such as Schaaf-Yang syndrome (*MAGEL2*), which had only one variant in our cohort, or *HNRNPH2*-related NDD, which had no variants in our cohort. Additionally, barriers to genetic testing likely impacted our cohort composition. Finally, we made the assumption that NDDs have similar prevalence globally, which is difficult to assess. While our estimates may reflect some ascertainment bias, these are still the most accurate estimates to date.

In addition to providing novel information for many RNDDs, this work also shows the values of exome or genome sequencing over panel analysis. While it is feasible to choose the top genes from our work for a panel, it is important to know that each of these affects 0.29% or less of individuals with NDDs. Thus, even with the top 100 genes, only 8.8% of potential RNDD diagnoses would be made. Even a panel of the top 500 genes would only have a diagnostic yield of <20%. In contrast, exome sequencing has an approximately 36% diagnostic yield and a higher yield for NDDs with comorbid conditions [69]. Our study supports the use of exome sequencing as a first-tier clinical diagnostic test for individuals with NDDs.

With this new approach to prevalence estimates, we hope that valuable information can be provided to families, clinicians, and groups developing potential therapeutics. Additionally, we show the value of large cohort studies in disease and emphasize the need for international collaboration. While these numbers are inherently in flux, we provide the most accurate prevalence estimates for many disorders to date.

## Figures and Tables

**Figure 1 biomedicines-10-02865-f001:**
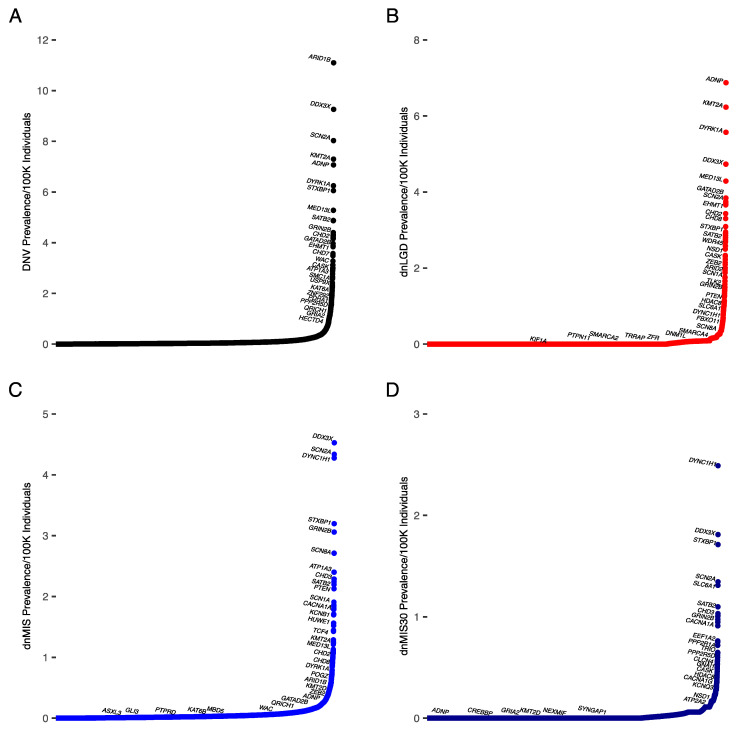
Prevalence of DNVs by gene extrapolated from percent of cases in total NDD cohort. Genes are indicated along the *x*-axis, with prevalence of each variant type on the *y*-axis. (**A**) NDD DNV cases, (**B**) NDD dnLGD cases, (**C**) NDD dnMIS cases, and (**D**) NDD dnMIS30 cases. The proportion of each gene and mutation type in our cohort was multiplied by the estimated prevalence of NDDs (DD/ID, ASD, and epilepsy) from Zablotsky and Black, 2020.

**Figure 2 biomedicines-10-02865-f002:**
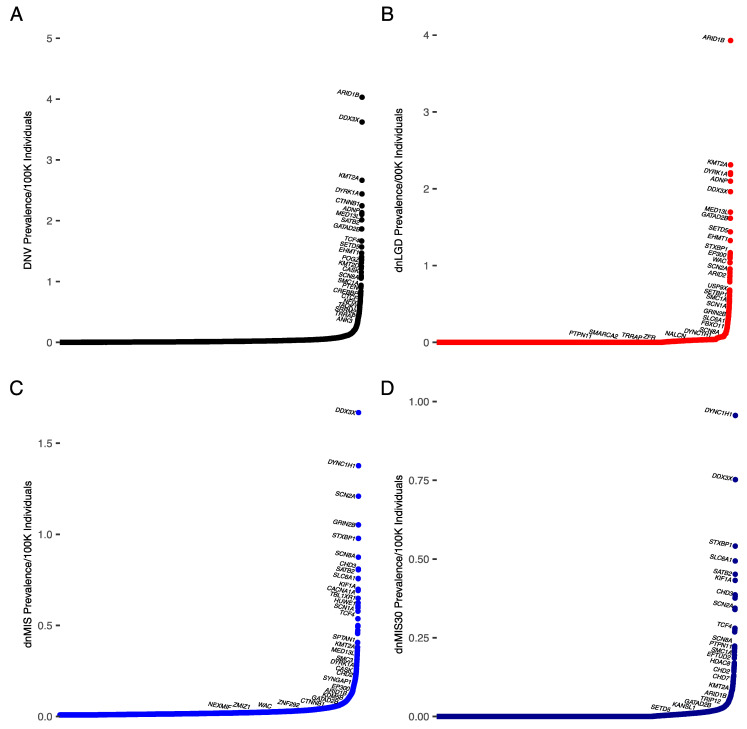
Prevalence of DNVs by gene extrapolated from number of cases in DD/ID cohort. Genes are indicated along the *x*-axis, with prevalence of each variant type on the *y*-axis. (**A**) DD/ID DNV cases, (**B**) DD/ID dnLGD cases, (**C**) DD/ID dnMIS cases, and (**D**) DD/ID dnMIS30 cases. The proportion of each gene and mutation type in our cohort was multiplied by the estimated prevalence of DD (DD/ID) from Zablotsky and Black, 2020.

**Figure 3 biomedicines-10-02865-f003:**
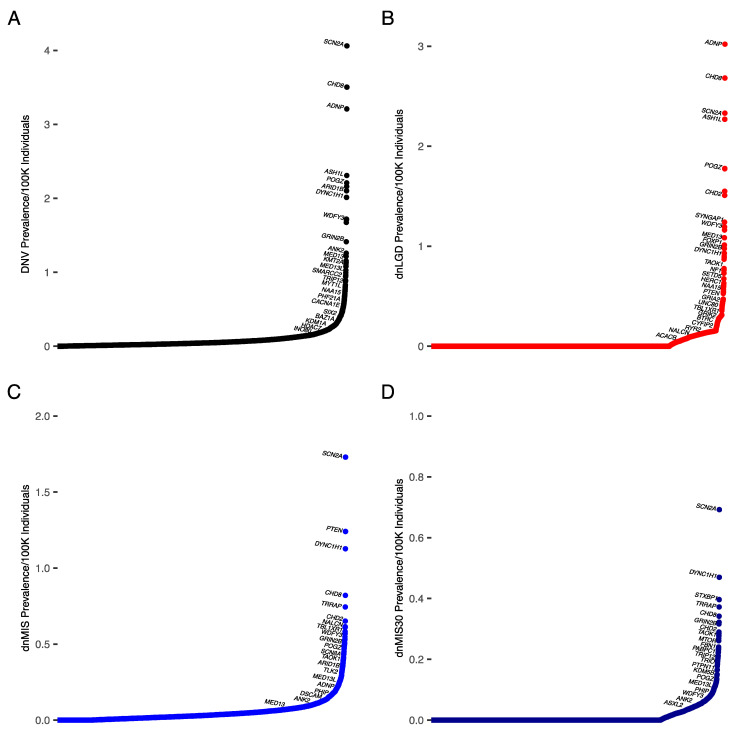
Prevalence of DNVs by gene extrapolated from number of cases in ASD cohort. Genes are indicated along the *x*-axis, with prevalence of each variant type on the *y*-axis. (**A**) ASD DNV cases, (**B**) ASD dnLGD cases, (**C**) ASD dnMIS cases, and (**D**) ASD dnMIS30 cases. The proportion of each gene and mutation type in our cohort was multiplied by the estimated prevalence of ASD from Zablotsky and Black, 2020.

**Figure 4 biomedicines-10-02865-f004:**
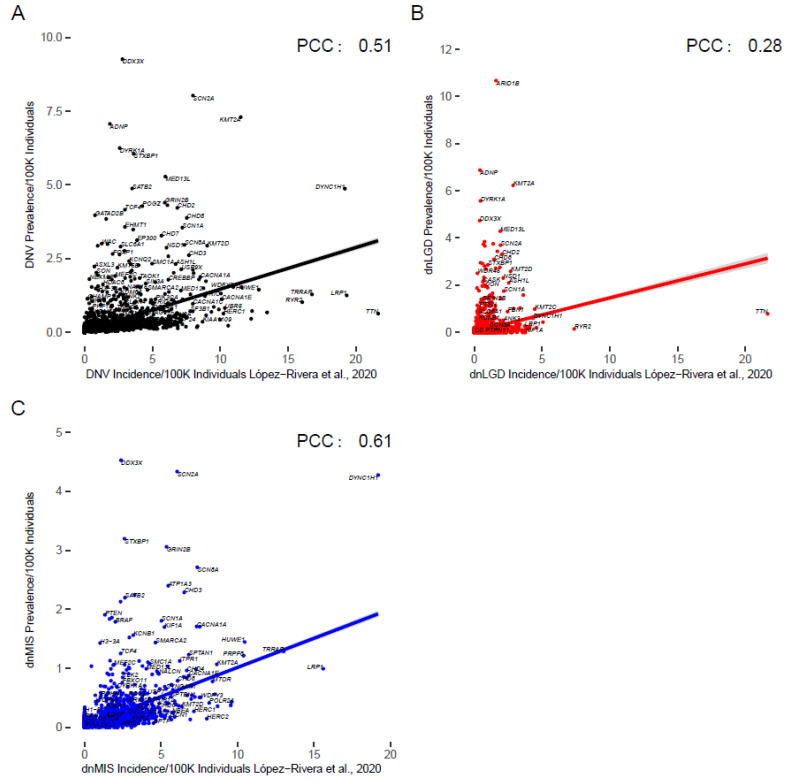
Prevalence of DNVs by gene versus incidence estimates from [14]. (**A**) NDD DNV cases (*p* < 0.0001 with Bonferroni correction), (**B**) NDD dnLGD cases (*p* < 0.0001), and (**C**) NDD dnMIS cases (*p* < 0.0001). All variant types had a positive correlation with previous incidence estimates, shown with Pearson’s correlation coefficients (PCC). Notably, some genes without clinical relevance, such as *TTN,* are also shown. Corrected *p* values and confidence intervals are shown in Appendix A.

**Table 1 biomedicines-10-02865-t001:** Prevalence estimates from Zablotsky and Black, 2020 of each disorder among 3- to 17-year old children in the US.

	Zablotsky and Black (2020) Prevalence %	95% Confidence Interval	Current Study n
**All NDDs**	4.5%	4–5%	50,377
**DD/ID**	1.2%	1.1–1.4%	31,191
**ASD**	2.5%	2.2–2.7%	16,125
**Epilepsy**	0.8%	0.7–0.9%	1389

**Table 2 biomedicines-10-02865-t002:** Top 10 most prevalent genes with variation among NDDs. Prevalence figures are normalized by constraint scores, thus may account a higher percentage of our cohort.

NDDs (Prevalence/100,000; % in Cohort)			
DNV	dnLGD	dnMIS	dnMIS30
*ARID1B* (11.1; 0.3%)	*ARID1B* (10.7; 0.25%)	*DDX3X* (4.5; 0.14%)	*DYNC1H1* (2.5; 0.09%)
*DDX3X* (9.3; 0.3%)	*ADNP* (6.9; 0.16%)	*SCN2A* (4.3; 0.17%)	*DDX3X* (1.8; 0.06%)
*SCN2A* (8; 0.3%)	*KMT2A* (6.2; 0.14%)	*DYNC1H1* (4.3; 0.16%)	*STXBP1* (1.7; 0.06%)
*KMT2A* (7.3; 0.22%)	*DYRK1A* (5.6; 0.13%)	*STXBP1* (3.2; 0.11%)	*SCN2A* (1.3; 0.05%)
*ADNP* (7.1; 0.17%)	*CTNNB1* (5.5, 0.13%)	*GRIN2B* (3.1; 0.13%)	*SLC6A1* (1.3; 0.05%)
*DYRK1A* (6.2.; 0.18%)	*DDX3X* (4.7; 0.11%)	*SCN8A* (2.7; 0.09%)	*SATB2* (1.1; 0.04%)
*STXBP1* (6.1; 0.23%)	*MED13L* (4.3; 0.1%)	*ATP1A3* (2.4; 0.08%)	*CHD3* (1; 0.05%)
*CTNNB1* (5.7; 0.13%)	*GATAD2B* (3.8; 0.09%)	*CHD3* (2.3, 0.1%)	*SMARCA4* (1; 0.04%)
*MED13L* (5.3; 0.18%)	*POGZ* (3.7; 0.09%)	*KCNQ2* (2.2, 0.1%)	*KIF1A* (1; 0.05%)
*SATB2* (4.9; 0.19%)	*SETD5* (3.7; 0.1%)	*SATB2* (2.2, 0.09%)	*GRIN2B* (1; 0.4%)

Values for all genes analyzed and their 95% confidence intervals are in Appendix A.

**Table 3 biomedicines-10-02865-t003:** Comparison of our prevalence estimates to previous estimates.

Gene/Syndrome	Current Cohort Estimate (All NDDs)	López-Rivera Estimate	Previous Estimates (Citation)
*ARID1B*/Coffin-Siris syndrome 1	Most are due to LGD variants: 1/9009	Most are due to LGD variants: 1/61,884	1/10,000–1/100,000 [47]
*EHMT1/KMT2C*/Kleefstra syndrome	LGD and MIS: *EHMT1:* 1/27,927 *KMT2C*: 1/90,759	LGD and MIS: 1/33,686 LGD and MIS: 1/22,373	At least 1/120,000 in those with NDDs [48]
*STXBP1/STXBP1* encephalopathy	LGD and MIS: 1/16,516	LGD and MIS: 1/27,664	1/91,862 [49]
*CHD7*/CHARGE syndrome	LGD and MIS: 1/30,513	LGD and MIS: 1/17,642	1/8500–1/15,000 newborns [50,51]
*KMT2D*/*KDM6A/*Kabuki syndrome	LGD and MIS: *KMT2D:* 1/34,054 *KDM6A:* 1/77,272	LGD and MIS: *KMT2D:* 1/11,061 *KDM6A*: 1/38,153	1/32,000–1/86,000 [52]
*NSD1*/Sotos syndrome	LGD and MIS: 1/34,843	LGD and MIS: 1/16,552.5	1/14,000 [53]
*ZEB2*/Mowat-Wilson syndrome	LGD and MIS: 1/43,906	LGD and MIS: 1/25,112	1/50,000–1/100,000 [54]
*MECP2*/Rett syndrome	LGD and MIS: 1/87,826	LGD and MIS: 1/486,085	1/10,000–1/23,000 female births [55]
*KANSL1*/Koolen-de Vries syndrome syndrome	LGD and MIS: 1/65,049	LGD and MIS: 1/59,802	May be as frequent as deletion (1/55,000) [56,57]
*SCN1A*/Dravet syndrome	LGD and MIS: 1/28,161	LGD and MIS: 1/13,877	1/22,000 incidence in Danish population [58]
*SLC2A1/*GLUT1 deficiency syndrome	MIS: 1/295,848	MIS: 1/58,766.5	1/33,898–1/83,333 [59,60]
*KCNQ2/KCNQ2* encephalopathy	MIS: 1/44,573	MIS: 1/30,534	1/84,746 [60]
*CREBBP/EP300*/Rubinstein-Taybi syndrome	LGD and MIS: *CREBBP*: 1/56,126 *EP300*: 1/32,028	LGD and MIS: *CREBBP*: 1/16,201 *EP300*: 1/25,862	1/100,000–1/125,000 [61]

**Table 4 biomedicines-10-02865-t004:** Top 10 most prevalent genes with variation among DD.

DD (Prevalence/100,000; % in Cohort)			
DNV	dnLGD	dnMIS	dnMIS30
*ARID1B* (4; 0.38%)	*ARID1B* (3.9; 0.2)	*DDX3X* (1.7; 0.2%)	*DYNC1H1* (1; 0.13%)
*DDX3X* (3.6; 0.37%)	*KMT2A* (2.3; 0.18%)	*DYNC1H1* (1.4; 0.2%)	*DDX3X* (0.8; 0.09%)
*KMT2A* (2.7; 0.28%)	*CTNNB1* (2.2; 0.18%)	*SCN2A* (1.2; 0.2%)	*STXBP1* (0.54; 0.07%)
*DYRK1A* (2.4; 0.25%)	*DYRK1A* (2.19; 0.17%)	*GRIN2B* (1; 0.18%)	*SLC6A1* (0.5; 0.07%)
*CTNNB1* (2.2; 0.2%)	*ADNP* (2.1, 0.16%)	*STXBP1* (0.96; 0.14%)	*SATB2* (0.5; 0.07%)
*ADNP* (2.1; 0.19%)	*DDX3X* (1.96; 0.13%)	*SCN8A* (0.9; 0.12%)	*KIF1A* (0.4; 0.08%)
*STXBP1* (2.1; 0.23%)	*MED13L* (1.7; 0.13%)	*KCNQ2* (0.8; 0.15%)	*CHD3* (0.4; 0.06%)
*SCN2A* (2.1; 0.28%)	*GATAD2B* (1.6; 0.12%)	*CHD3* (0.8; 0.14%)	*ATP1A3* (0.4; 0.04%)
*MED13L* (2; 0.24%)	*SETD5* (1.4; 0.12%)	*SATB2* (0.7; 0.12%)	*CACNA1A* (0.4; 0.07%)
*SATB2* (1.9; 0.21%)	*EHMT1* (1.3; 0.11%)	*ATP1A3* (0.7; 0.1%)	*SMARCA4* (0.3; 0.05%)

Values for all genes analyzed and their 95% confidence intervals are in Appendix A.

**Table 5 biomedicines-10-02865-t005:** Top 10 most prevalent genes with variation among ASD.

ASD (Prevalence/100,000; % in Cohort)			
DNV	dnLGD	dnMIS	dnMIS30
*SCN2A* (4.1; 0.22%)	*ADNP* (3, 0.12%)	*SCN2A* (1.7; 0.12%)	*SCN2A* (0.7; 0.05%)
*CHD8* (3.5, 0.19%)	*CHD8* (2.7; 0.11%)	*PTEN* (1.2; 0.07%)	*DYNC1H1* (0.5; 0.03%)
*ADNP* (3.2; 0.16%)	*SCN2A* (2.3; 0.1%)	*DYNC1H1* (1.1; 0.07%)	*STXBP1* (0.4; 0.03%)
*ASH1L* (2.3; 0.1%)	*ASH1L* (2.3; 0.09%)	*CHD8* (0.8; 0.07%)	*TRRAP* (0.4, 0.03%)
*POGZ* (2.2; 0.12%)	*ARID1B* (1.8; 0.07%)	*TRRAP* (0.7; 0.06%)	*CHD8* (0.3; 0.03%)
*CHD2* (2.2; 0.12%)	*POGZ* (1.8; 0.07%)	*CHD2* (0.65; 0.06%)	*GRIN2B* (0.3; 0.03%)
*ARID1B* (2.1; 0.14%)	*KMT5B* (1.6; 0.06%)	*NALCN* (0.6; 0.06%)	*CLCN4* (0.3; 0.02%)
*DYNC1H1* (2; 0.11%)	*CHD2* (1.5; 0.06%)	*PABPC1* (0.6; 0.04%)	*CHD2* (0.3; 0.02%)
*WDFY3* (1.7; 0.1%)	*SYNGAP1* (1.2; 0.05%)	*CYFIP2* (0.58; 0.04%)	*SLC6A1* (0.28; 0.02%)
*KMT5B* (1.7; 0.08%)	*WDFY3* (1.2; 0.05%)	*TBL1XR1* (0.56; 0.03%)	*SMARCA4* (0.27; 0.02%)

Values for all genes analyzed and their 95% confidence intervals are in Appendix A.

## Data Availability

The data presented in this study are available in the Appendix A.

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
