# Peer review of "Estimating the Prevalence of De Novo Monogenic Neurodevelopmental Disorders from Large Cohort Studies"

_biomedicines, 2022, doi:10.3390/biomedicines10112865_

Round 1

Reviewer 1 Report

The authors reported the prevalence of de novo monogenic neurodevelopmental disorders from large cohort studies which is valuable on its own. The research in interesting. The authors have presented many information pertaining to their work, however, there are some major questions and minor revisions need to improve the quality of some figures and English language wrting.

- I asuume that the data were extracted from already published genome and exome studies worldwide and newly generated data. If this is true, I suggest to revise the sentense “Rare diseases impact up to 30 million Americans” and reports the real estimation of people from whole world not only the Americans. Otherwise, the current research will be considered as a local study. Furthermore, could you find any correlation between the genes, disorders, and the frequency in different populations?

- Developmental delay, autism spectrum disorder, and epilepsy are complex disorders affected by many factors included genetics. Several studies have reported a multi gene panel for such disorders. What is the novelty of your research? Any novel genes or variant or more efficient modified panel? Please clarify.

- Although the authors reported the prevalence of de novo monogenic neurodevelopmental disorders from large cohort studies, however, this assay is not validated experimentaly. How do you confirm that the related phenotypes are related to genes or due to a point mutations?

- There are some minor writing errors should be revised:

“We find that the prevalence of many disorders is positively correlated to the previously estimated incidence”. It is better to say “We found …”. Etc.

- Figure S2 can be presented in a more coherent format.

Author Response

We have listed below the comments and suggestions from Reviewer 1, followed by our response in blue:

  1. “I assume that the data were extracted from already published genome and exome studies worldwide and newly generated data. If this is true, I suggest to revise the sentence “Rare diseases impact up to 30 million Americans” and reports the real estimation of people from whole world not only the Americans. Otherwise, the current research will be considered as a local study. Furthermore, could you find any correlation between the genes, disorders, and the frequency in different populations?”

To address this, we updated the abstract to say that 400 million individuals are diagnosed with rare disease globally. As these are all de novo variants, there should not be any differences in frequency in various populations, for the most part, as these are randomly occurring.

  1. “What is the novelty of your research? Any novel genes or variant or more efficient modified panel? Please clarify.”

This work is not focusing on identifying new genes, but the prevalence of variation in genes among patient populations. So, while the data itself is not novel, the use of the data to determine prevalence is.

  1. “Although the authors reported the prevalence of de novo monogenic neurodevelopmental disorders from large cohort studies, however, this assay is not validated experimentally. How do you confirm that the related phenotypes are related to genes or due to a point mutations?”

These cohorts are all clinically affected individuals, and some have had variants reported back through their various studies. A small subset has been Sanger validated. As these variants were identified using stringent quality control metrics, and some have been called using multiple pipelines, we expect the majority of the variants to be real. They are primarily point mutations in genes (as well as indels, but no large copy number variants were assessed).

Reviewer 2 Report

It is an important data regarding rare disease, particularly NDD. I have several suggestions:

- please focus on NDD, including the Introduction section.

- please reorganize the manuscript into systematic review that narrative review.

- I suggest to omit the social media part. It distracts the message of the main findings.

Author Response

We have listed the comments and suggestions from Reviewer 2 and followed with our response in blue:

  1. asked for focus on NDDs and reorganization of the manuscript.

We have added focus to NDDs primarily in the introduction, but throughout the manuscript as well, with the only references to broader rare disease being in the abstract and early introduction. We have also clarified NDDs (broader diagnoses, ex: DD/ID or ASD) and rare NDDs (RNDDs, genetic diagnosed NDDs). 

  1. They also suggested removing the social media information from the manuscript.

We have chosen to keep this information as this may be a valuable tool epidemiologically.

Reviewer 3 Report

Clinicians and epidemiologists are aware of the high frequency of de novo mutations (NM) in autism spectrum disorder (ASD). Several publications had attempted an estimation of the frequency of these mutations. In the present MS the authors group several previously published analyses reaching a high number of patients (50,377) and many coding genes (11.000). Several weighting strategies improve the estimation of NM. The study is conducted with care (the authors have cautiously eliminated overlapping patients). The comparison of the new results with those resulting from previous publications shows the adequacy of the methods and a good reliability of the NM. 

I have several remarks or questions about the MS. 

1)     How was made the diagnoses? What were the criteria? Did the studies used the DSM-5? It is impossible to publish a paper without indicating how the identification of the pathologies were done. 

2)    Why the Bonferroni correction that generates false negatives? Why not false discovery rate (FDR), with the Benjamini-Hochberg (BH) procedure?

3)    With your analysis the crucial results are not provided by the inferential statistics but by the value of the effect size. Under the condition you should discuss the part of common variance and not its square root. In other words, you should not discuss Bravais-Pearson r but r2

4)    No indication for most of the axis in the figures. 

5)    What is figure S2 in the MS?

6)   People out of USA have ASD and people perform valuable research on ASD, out of USA . Some epidemiological research is done with ASD elsewhere. It could be loyal to examine these results. Please correct  this part of the introduction. 

Author Response

We have listed Reviewer 3's suggestions and comments and followed with our responses in blue:

  1. How was made the diagnoses? What were the criteria? Did the studies used the DSM-5? It is impossible to publish a paper without indicating how the identification of the pathologies were done. 

These studies are well-established cohorts of affected individuals, with varying diagnoses. Exact diagnostic criteria are available for some, such as SSC in SFARI Base. While we do not have the exact DSM-5 diagnoses, the phenotypes of each cohort are listed in Table S1. We have also provided all available diagnostic criteria in Table S1.

  1. Why the Bonferroni correction that generates false negatives? Why not false discovery rate (FDR), with the Benjamini-Hochberg (BH) procedure?

We have updated Table S6 to have both Bonferroni and FDR p values, which are equal due to the large sample size used (50,377 probands * 20,000 genes).

  1. With your analysis the crucial results are not provided by the inferential statistics but by the value of the effect size. Under the condition you should discuss the part of common variance and not its square root. In other words, you should not discuss Bravais-Pearson r but r2

We believe the use of R is appropriate as we are showing that the prevalence values we calculated and the incidence values calculated in previous studies are similar, not that one influences the other. We are also not looking at the amount of variance as a result of a variant, we are assuming these are autosomal dominant with full penetrance.

  1. No indication for most of the axis in the figures. 

We added what the axes designations in the figure legends.

  1. What is figure S2 in the MS?

We added the figure legend for S2.

  1. People out of USA have ASD and people perform valuable research on ASD, out of USA. Some epidemiological research is done with ASD elsewhere. It could be loyal to examine these results. Please correct this part of the introduction. 

We have considered the global DD/ASD numbers. Actually, our ASD estimate is higher than recently published global prevalence (0.4%-1.7% based on region, Salari et al., 2022). As these are de novo, random variants, we assume that the prevalence is similar globally. We are making the assumption that ASD and DD/ID occur at similar rates globally. We added a sentence to note our assumption in the discussion:

“Finally, we have made the assumption that NDDs have similar prevalence globally, which is difficult to assess.”

Round 2

Reviewer 1 Report

Figure 2S can be presented in a more compact format.  The figure occupies approximately 3/4 of the page. I suggest editing the figure.  

Reviewer 3 Report

OK but we do not know what was the exact diagnostic.